# Metamorphic Effect of Angiogenic Switch in Tumor Development: Conundrum of Tumor Angiogenesis Toward Progression and Metastatic Potential

**DOI:** 10.3390/biomedicines11082142

**Published:** 2023-07-29

**Authors:** Daniel Leon Moshe, Leili Baghaie, Fleur Leroy, Elizabeth Skapinker, Myron R. Szewczuk

**Affiliations:** 1Faculty of Health Sciences, Queen’s University, Kingston, ON K7L 3N9, Canada; daniel.moshe@queensu.ca; 2Department of Biomedical & Molecular Sciences, Queen’s University, Kingston, ON K7L 3N6, Canada; 16lbn1@queensu.ca; 3Faculté de médecine, Maïeutique et Sciences de la Santé, Université de Strasbourg, F-67000 Strasbourg, France; fleur.leroy@etu.unistra.fr; 4Faculty of Arts and Science, Queen’s University, Kingston, ON K7L 3N9, Canada; 21ess18@queensu.ca

**Keywords:** angiogenesis, anti-angiogenic treatment, control systems, tumor microenvironment

## Abstract

Our understanding of angiogenesis has significantly expanded over the past five decades. More recently, research has focused on this process at a more molecular level, looking at it through the signaling pathways that activate it and its non-direct downstream effects. This review discusses current findings in molecular angiogenesis, focusing on its impact on the immune system. Moreover, the impairment of this process in cancer progression and metastasis is highlighted, and current anti-angiogenic treatments and their effects on tumor growth are discussed.

## 1. Introduction

### The Vital Role of Angiogenic Growth Factors in Cancer Progression

Angiogenesis is seen as the process that mediates new blood vessel formation and capillaries, an essential process allowing for the exchange of nutrients throughout the body [1,2,3]. Many signaling pathways contribute to angiogenesis; however, vascular endothelial growth factor (VEGF) is recognized as the most influential of the angiogenic growth factors (AFs) [4]. When VEGF binds to one of its receptors (VEGFRs), it triggers a signaling cascade leading to vasculogenesis and angiogenesis [5]. While required for normal physiological functions, the signaling pathway is also implicated in cancer progression when dysregulated. For tumors to proliferate, they require nutrient delivery via neovascularization; therefore, these AFs are vital for cancer progression [2]. Previous findings have shown that VEGFRs are abundantly expressed in both liquid and solid tumor cells, and the binding of VEGF to its receptor on tumor cells is associated with the activation of the MAPK signaling cascade leading to tumor invasion and metastasis [6,7]. Moreover, hypoxia, which is a staple of the tumor microenvironment (TME), induces VEGF and VEGFRs through the hypoxia-inducible factor (HIF) [1,3,8].

Like VEGF, the placental growth factor (PIGF) has pro-angiogenic properties; however, unlike the direct effect of VEGF, PIGF synergizes with VEGF-A to bind to VEGFR on monocytes, triggering the release of inflammatory cytokines TNF-α and IL-6 [1,5]. By releasing these cytokines, PIGF modulates the innate immune system by activating macrophages and attracting monocytes leading to an increased inflammatory state in wound healing and cancer [1]. Like VEGF, PIGR and its coreceptor neuropilin-1 (NRP1) are upregulated in hypoxic environments such as the TME [1]. Other stimuli, such as oncogenes and hormones, are also suggested to upregulate PIGF, making their presence in tumor cells and the tumor microenvironment all the more likely [1]. Until recently, these growth factors were studied individually; however, recent reports have discussed their interconnectivity. For example, PIGF has been shown to upregulate VEGFR-1 as well as other pro-angiogenic growth factors like platelet-derived growth factor β (PDGF-β), fibroblast growth factor 2 (FGF-2), and matrix metalloproteinases (MMPs) [1,9].

Another set of angiogenic growth factors called fibroblast growth factors (FGFs) are implicated in both angiogenesis and tumor progression. Under homeostasis, FGF is involved in the development of vascularization to skeletal and nervous tissues and wound healing [2]. When FGFs bind to their receptors (FGFRs), it triggers downstream signaling of the PI3K/AKT, RAS/MAPK, and JAK/STAT pathways [10]. FGFRs are receptor tyrosine kinases (RTKs), and when these proteins are dysregulated or overexpressed, there is an increase in malignant diseases such as cancer [2,11]. Increased levels of FGFRs are found in gliomas, and higher levels of FGF-2 are correlated to high-grade tumors [2]. Like PIGF, FGF can upregulate the expression of VEGF, further demonstrating the interconnectedness of these growth factors [2].

Platelet-derived growth factors (PDGFs) are also linked with the regulation of angiogenesis by activating VEGF-expressing stromal tumor-associated fibroblasts [6]. Like the previously discussed growth factors, PDGF is involved in the development of the vascular system and is directly linked to central nervous system (CNS) development and wound healing [2]. Its receptors (PDGFRs) are also RTKs, and high expression of PDGFs corresponds to higher tumor grades [2]. Since PDGFs are primarily present on endothelial cells at the tip of sprouting vessels, environmental changes such as hypoxia activate PDGF and induce the migration of those endothelial cells to the tumor tissue to create new vascularization that benefits the growth of the tumor [12,13].

A growth factor implicated in angiogenesis but far less focused on compared to the AFs discussed thus far is the transforming growth factor beta (TGFβ). TGFβ can exert its angiogenic functions through the upregulation of VEGF and activation of the MAPK pathway, and the high expression of TGFβ in the TME is correlated with poorer patient outcomes [14].

## 2. Contribution of Angiogenic Growth Factors in Immunosuppression and Immune Escape

AFs influence both innate and adaptive immune cell populations within the TME to create a more tolerogenic milieu. Myeloid regulatory cells such as myeloid-derived suppressor cells (MDSCs), tumor-associated macrophages (TAMs), type 2 natural killer T (NKT) cells, and regulatory T-cells (Tregs) are the primary cells types that contribute to immune escape and immunosuppression within the tumor microenvironment [15,16].

### 2.1. Tumor-Associated Macrophages and Angiogenic Growth Factors

Tumor-associated macrophages differentiate into anti-inflammatory M1 macrophages (M1-TAMS) or tumor-promoting and proinflammatory M2 macrophages (M2-TAMS) [17,18]. The polarization of TAMs into their respective phenotypes is dependent on local signals provided in the tumor milieu [17]. VEGF, directly and indirectly, polarizes monocytes towards an M2 phenotype, contributing to an immunosuppressive tumor microenvironment [19]. M2-TAMS express cytokines IL-6, IL-10, CCl-22, and TGF-β, which, while promoting monocyte maturation, block the differentiation of monocytes to dendritic cells and disrupt the activity of cytotoxic T lymphocytes (CTLs) and NK cells [20,21]. It was previously understood that VEGF induces monocyte recruitment to the tumor site by stimulating endothelial cells to release monocyte chemoattractant protein (MCP-1), increasing the endothelial layer’s permeability to enhance cell migration [22]. Monocytes would thus have increased opportunity for direct contact with tumor cells and exposure to the various signals present in the tumor microenvironment, skewing monocytes to differentiate into an M2-TAM phenotype [17] (Figure 1). 

PIGF binding to VEGFR-1 stimulates the recruitment of macrophages into the tumor microenvironment [23,24,25,26,27]. PIGF is upregulated in many advanced stages of cancer and may also play a role in monocyte recruitment, separate from its stimulation of VEGF secretion. Previously, it was established that factors like VEGF could influence M2-TAMs and promote their polarization towards the M2 phenotype associated with tumor-promoting functions. However, recent data suggest that in the context of M2-TAM polarization, the signaling of PlGF might play a more critical role than VEGF signaling through VEGFR-1. Macrophage polarization and its association with PlGF have been linked to Histidine-rich glycoprotein (HRG), a plasma protein with anti-inflammatory effects [21]. By binding to different cells, HRG, produced by monocytes and macrophages, can modulate various functions, including immunity and vascularization. HRG exhibits both anti-angiogenic and pro-angiogenic activity, which is believed to be due to its disruption of the endothelial cell cytoskeleton, inhibiting vessel formation. [21]. In contrast, HRG’s pro-angiogenic activity results from its high-affinity binding to thrombospondin-1 (TSP-1), masking TSP-1′s anti-angiogenic epitope [26]. The inhibition of angiogenesis by TSP-1 and HRG depends on TSP-1 interacting with CD47, which not only inhibits endothelial cell proliferation but also downstream eNOS/NO/cGMP signaling and VEGFR-2 phosphorylation [28]. Importantly, this inhibition of VEGFR-2 phosphorylation does not prevent the binding of VEGF-VEGFR [28].

Studies have shown that overexpression of HRG in specific cancer cells leads to slower tumor growth and reduced metastasis in mice, even with persistent TAMs accumulation [29]. HRG exposure downregulates M2 markers in TAMs, such as IL10, CCL22, and PlGF, while upregulating M1 markers, such as IL6 and CXCL9 [23]. This shift in TAM polarization helps decrease the immunosuppression within the tumor microenvironment by decreasing regulatory T-cell (Treg) recruitment, improving DCs and T-cells function, and promoting infiltration of CD8^+^ T-cells and NK cells into the tumor stroma [23].

HRG’s effect on TAMs polarization appears to be modulated mainly by the downregulation of PlGF. Without host derived PlGF, HRG does not further suppress tumor growth. One study showed that PlGF expressed by non-small cell lung cancer (NSCLC) cells triggers TAM polarization and promotes tumor growth and metastasis [29]. In breast cancer (BC) and murine pancreatic ductal adenocarcinoma (PDAC) models implanted in obese mice, targeting PlGF/VEGFR-1 signaling led to a shift in the profile of tumor secreted cytokines and TAMs differentiation towards the M1 phenotype, resulting in reduced tumor progression [30]. It was found that plasma PlGF was associated with obesity in PDAC and BC patient samples and that VEGF-A was not, further supporting the role of PlGF in TAM polarization towards the M2 phenotype [30]. In a more recent study, Ma et al. [31] demonstrated that using metformin (200 mg/kg day) was enough to repolarize M2-TAMs into M1-TAMs and to decrease tumor progression even in the presence of PIGF autocrine signaling. Another study found that an inhibitor of HIF-1α, Lificiguat (YC-1), in triple-negative breast cancer (TNBC), also had similar results in repolarizing TAMs and inhibiting angiogenesis and tumor growth [32].

These findings suggest that PlGF, rather than VEGF through VEGFR-1, is a crucial driver of TAM polarization towards the immunosuppressive M2 phenotype [21]. This highlights the potential importance of targeting PlGF signaling as a therapeutic strategy to modulate TAM polarization and promote an anti-tumoral immune response within the tumor microenvironment. It is also essential to recognize that many factors, such as FGF, are highly associated with M2 polarization, whose mechanisms are yet to be fully elucidated [10]. Further research is required to understand the role of various AFs across tumor heterogeneity.

### 2.2. Myeloid-Derived Suppressor Cells (MDSCs) and Angiogenic Growth Factors

M2 polarized macrophages secrete Th2 cytokines, such as IL-10, TGF-β, CCL-22, IL-6, and growth factors, such as VEGF and PIGF, contributing to angiogenic remodeling [17]. VEGF, IL-6, CCL-22, and IL-10 are also linked to further stimulation and recruitment of MDSCs to the TME [33,34]. MDSCs contribute to a more immunosuppressive tumor milieu by secreting anti-inflammatory cytokines such as IL-10 and TGF-β [35]. TGF-β contributes to the induction and production of Tregs into the TME, and IL-10 can arrest the production of interferon-γ (IFN-γ) by CD4^+^ T-cells, which further drives cancer progression and metastasis [36]. MDSCs, as an immature and undifferentiated population, can further differentiate into macrophages such as M2-TAMs, and DCs [36]. MDSCs, as immature myeloid cells, are well known to depress the T-cell function. Immune checkpoint regulators, such as the programmed-death ligand 1 (PD-L1), are known to be expressed on MDSCs [36]. HIF-1α regulates PD-L1 gene expression. Thus, under hypoxic conditions such as the environment within the TME, PD-L1 is overexpressed by MDSCs [37]. The programmed death 1 receptor (PD-1) to which PD-L1 binds is expressed by the effector CD4^+^ and CD8^+^ T-cells, DCs, and APCs [36]. PD-L1 on MDSCs interact with T-cells which express PD-1, resulting in T-cell anergy, where there is diminished cytokine production [36]. Additionally, MDSCs carry exosomal cargo rich in cytokines and AFs, which drive cancer progression and metastasis when released into the TME [36,38]. MDSCs use abundant VEGF in the TME and bind to VEGFR to initiate signaling cascades involving JAK2/STAT3 to produce additional angiogenic molecules [36]. Interestingly, it was found that the activation of MDSCs by VEGF resulted in increased immunosuppressive activity relative to non-exposed MDSCs [39]. Stimulation of MDSCs by proinflammatory cytokines and VEGF allows the production of VEGF through a STAT-3-mediated pathway. This positive feedback loop recruits further MDSCs to the TME and results in the further release of AFs [36].

### 2.3. Dendritic Cells, T-Cells, Tregs Signaling, and Angiogenic Growth Factors

Dendritic cells (DCs) are crucial in activating T-cells through antigen presentation. However, various AFs have been found to inhibit dendritic cell maturation and antigen presentation. Recent work has demonstrated an association between high levels of VEGF expression in human cancers, impaired cell function, and a reduced number of cells [40]. Recall that VEGF, PIGF, and FGF are involved in the recruitment of cells such as M2-TAMs and MDSC, which can block the differentiation of DC and increase PD-L1 expression within the TME. VEGF upregulates PD-L1 in dendritic cells, inhibiting T-cell expansion and function, and inhibits the function of mature dendritic cells to stimulate T-cells by acting on VEGFR-2 and inhibiting NF-κB activation [41,42]. It also interferes with the ability of mature dendritic cells to stimulate T-cells through the involvement of VEGFR-2. FGF/FGFR signaling through the JAK/STAT pathway has also been associated with increased expression of PD-L1 through the upregulation of Yes-associated protein (YAP) [10,41]. YAP is considered to have an oncogenic role, aiding in inhibiting tumor apoptosis and triggering metastasis and therapeutic evasion [43]. PIGF is also implicated in the suppression of DCs and inhibition of T-cell proliferation through the binding of PIGF to VEGFR-1 [44,45]. Moreover, PIGF has been found to upregulate the secretion of anti-inflammatory cytokines such as IL-10 by CD4^+^ and CD8^+^ T-cells leading to increased immunosuppression [44].

VEGF contributes to CD8^+^ T-cell exhaustion through VEGFR-2 and activated T-cell nuclear factor-mediated process [41,46]. CD8^+^ T-cell exhaustion can occur due to the expression of negative immune checkpoints such as PD-1, CTLA-4, T-cell immunoglobulin mucin receptor 3, and lymphocyte activation gene 3 protein [41,46]. Furthermore, the induction of FAS-ligand expression on endothelial cells by VEGF establishes a selective immune barrier that can suppress effector T-cell functions and cause apoptosis of CD8^+^ T-cells [47]. VEGF also promotes Treg generation, impeding CD8^+^ and CD4^+^ T-cell differentiation within the thymus [41]. VEGF can function as a chemoattractant in recruiting FOXP3+ Treg cells, suppressing the anti-tumor response [41,42]. VEGF signaling through VEGFR-2 also aids in the induction and survival of Tregs within the TME [42]. FGF/FGFR signaling has also increased the survival of Tregs through IL-2-mediated STAT5 phosphorylation [10].

## 3. The Impact of Angiogenic Growth Factors on Metastasis

Metastasis is overwhelmingly responsible for poor patient outcomes and cancer-related deaths, linked to 90% of breast cancer deaths, 35% of colorectal, and 85% of pancreatic cancer cases [48,49,50]. Once cancer metastasizes, it becomes increasingly difficult to treat as its extensive spread to secondary organs occurs rapidly. There is increasing evidence that residual cancer cells can easily transition into cancer stem cells (CSCs) which are responsible for metastasis through epithelial-mesenchymal transformation (EMT) [51]. This process results in a phenotypic change in which the loss of the tight cell-cell junctions and cellular polarity on epithelial cells leads to their detachment and migration to distant organs through circulation [52,53,54,55]. The result is tumor cells with increased proliferation, invasion, metastasis, and the ability to self-renew [52]. These mesenchymal cells also have enhanced resistance to apoptosis, which tumor cells use to evade immune detection and degradation [52]. EMT is triggered through various signals from the tumor microenvironment, such as those induced by growth factors and cytokines.

### 3.1. Role of Angiogenic Growth Factors in Cell Detachment

Metastasis occurs in three essential steps: the detached tumor cells leave the primary tumor, enter circulation, and finally travel to distant organs, where they proliferate, creating secondary lesions [56]. Therefore, the extensive vasculature network due to angiogenesis creates vessels and capillaries easily penetrable by tumor cells [8]. VEGF and FGF signaling by tumor cells is the key to the first step of detachment from the basement membrane and entrance to circulation [8]. PLGF also plays a role in this step as it activates MMP-9, which breaks down the extracellular matrix, via the p38-MAPK signaling pathway [57]. Moreover, the overexpression of MMP-9 and other members of the metalloproteinases family, including MMP-7, contribute to the metastatic spread of cancer [57,58].

### 3.2. Angiogenic Growth Factors in Autocrine Signaling

VEGFRs on tumor cells have a documented role in autocrine signaling which is involved in inflammation and metastasis [6]. VEGFR-2, for example, is known to form a complex with the receptor tyrosine kinase MET, whose ligand is the hepatocyte growth factor (HGF). HGF signaling is vital in stimulating the invasive properties of tumor cells through EMT [59,60]. Moreover, the coreceptors NRP1 and NRP2, which interact with VEGF, also bind to TGFβ [59]. TGFβ has many functions depending on the target; it triggers angiogenesis when interacting with VEGF, PDGF, MMP-9, and EMT when it binds to Snail1 [61]. In fact, under hypoxic conditions, TGFβ gains a motility function that directly impacts cellular invasion [61].

Hypoxia is an essential factor in stimulating angiogenesis. However, it also increases the expression of EMT markers like N-cadherin, Snail, fibronectin, and vimentin while downregulating the epithelial marker E-cadherin [3,52,54,56,62,63,64,65].

## 4. Anti-Angiogenic Therapy

Anti-angiogenic therapy can be approached in many ways (Table 1). Approaches can involve using a drug that targets the pathways involved in angiogenesis or blocking the activity of specific molecules that promote blood vessel formation. An example of the former would be oseltamivir phosphate monotherapy, which disrupts angiogenesis and EMT through interference with neuraminidase-1 (Neu1) activity following epidermal growth factor’s (EGF) stimulation [66,67]. Oseltamivir phosphate monotherapy impeded neovascularization, growth, and metastasis in a mouse model of human pancreatic carcinoma, triple-negative breast adenocarcinoma, and human ovarian carcinoma [66,67,68]. The expressions of Snail and MMP-9 are closely connected in similar invasive tumor processes [68]. Here, Snail induces MMP-9 secretion using signaling pathways, particularly the cooperation with oncogenic H-Ras (RasV12), leading to the transcriptional up-regulation of the *mmp9* gene. This Snail-MMP-9 signaling promotes glycosylation modification of growth factor receptors, involving the enzyme crosstalk Neu1-MMP-9 tethered at the ectodomain of RTKs. Activated MMP-9 is proposed to remove the elastin-binding protein (EBP) as part of the molecular multi-enzymatic complex that contains β-galactosidase/Neu1 and protective protein cathepsin A (PPCA) [68]. Activated Neu1 hydrolyzes α-2,3-sialic acid residues of RTKs at the ectodomain to remove steric hindrance to receptor association and activation. This process sets the stage for Snail’s role in tumor neovascularization. Abdulkhalek et al. [68] provided evidence showing the ability of transcription factor Snail to mediate ovarian tumor neovascularization. The silencing transcriptional factor Snail in A2780 ovarian carcinoma cells ablated the abnormal robust and bloody tumor vascularization in RAGxCγ double-mutant mice with a concomitant abolishment of tumor growth and metastatic spreading to the lungs.

Moreover, it has been shown that Snail induces MMP-9 secretion via multiple signaling pathways, but particularly in cooperation with oncogenic H-Ras (RasV12), Snail leads to the transcriptional upregulation of MMP-9 [69]. Collectively, these different signaling paradigms involved with EMT in ovarian cancer suggest that growth factor receptor glycosylation modification involving the receptor-signaling platform of a Neu1-MMP-9 crosstalk may be the invisible link connecting the Snail-MMP-9 signaling axis. It follows that the therapeutic efficacy of oseltamivir phosphate targeting Neu1 tethered to these receptors would be critically dose-dependent (Figure 2).

In preclinical xenografts of pancreatic cancer, the profiles of multiple phosphorylated proteins in the tumor lysates involved in cell signaling pathways were investigated. Individual tumors taken from the oseltamivir phosphate (OP) treated cohorts expressed significantly less phosphorylation of EGFR-Tyr1173, Stat1-Tyr701, and NFκBp65-Ser311 compared to the untreated cohort as determined by protein analysis or protein expression [70]. The Bio-Plex multiplex format also showed a reduction in phosphorylation of Akt-Thr308, PDGFRα-Tyr754, and STAT1-Tyr701. However, unexpectedly, there was an increase in phospho-Smad2-Ser465/467 and phospho-VEGFR-2-Tyr1175 in the tumor lysates from the OP-treated cohort compared to the untreated cohort. To explain these latter findings, Gilmour et al. [70] proposed that the increased phospho-VEGFR-2-Tyr1175 expression in tumor lysates from OP-treated cohorts could be due to prolonged VEGF signaling in the absence of endothelial epsins 1 and 2 (*Epn1/2*) producing leaky defective tumor angiogenesis, and thus it contributes to tumor growth retardation. Epsins, a family of ubiquitin-binding endocytic clathrin adaptor proteins, play an essential role in regulating angiogenesis by preventing the interaction of VEGF and VEGFR under normal physiological conditions [71]. Paradoxically, in tumor cells, the knockout of epsins 1 and 2 results in decreased tumor growth by increased VEGF signaling. Pasula et al. explain this phenomenon by the excessive VEGF signaling when negative regulation by epsins is blocked [71]. In their study, Epn1/2 knockout mice exhibited highly disorganized vascular structures with increased vascular permeability in tumors due to increased VEGFR-2 signaling, non-productive tumor angiogenesis, and retarded tumor growth. Indeed, tumor vasculature requires stringently balanced VEGF signaling to provide sufficient productive angiogenesis for tumor development. The findings in Gilmour’s report [70] indicated that OP treatment of tumor-bearing mice may have disrupted tumor vasculature by blocking epsins via an unknown mechanism. Their data suggested a potential anti-angiogenic strategy by which OP treatment of tumor-bearing mice may down-regulate endothelial Epn1/2 functions and promote local excessive VEGF signaling. This novel pro-VEGFR-2 and pro-Smad2 signaling as a consequence of OP therapy proposes another anti-cancer role for OP therapy with an uncharacterized property with broader specificities than expected. These findings highlight an excellent opportunity to target anti-VEGF-resistant tumors by targeting epsins 1 and 2.

Anti-angiogenesis inhibitors can also cause a temporary decrease in tumor hypoxia, as VEGF temporarily normalizes the function of tumor-associated vasculature by decreasing vascular permeability and restoring sustained pressure gradients, increasing O2 delivery and perfusion of cytotoxic agents to the tumor site [72]. However, this vasculature normalization is a transient process that only occurs at the beginning of treatment. Tumor blood vessels lose their maturation, and hypoxia increases during prolonged VEGF inhibition [73]. This hypoxia results in the secretion of angiogenic cytokines and thus an increase of alternative angiogenic pathways such as FGF, PDGF, PIGF, TNF-α, and SDF-1α, which can reduce the efficacy of anti-VEGF inhibitors [10,74]. Anti-angiogenic resistance has also been associated with M2-TAMs, specifically, PIGF-mediated M2TAM recruitment [25,75].

Another example of anti-angiogenic therapy would be pentoxifylline, a synthetic xanthine derivative that acts as a phosphodiesterase inhibitor. A growing body of evidence demonstrates that pentoxifylline can block the release of inflammatory cytokines in stimulating AFs secretion in a dose-dependent manner [76,77].

The most common approach to anti-angiogenic therapy is through the blockage of VEGFRs or ligands by antibodies and inhibiting RTK enzymes [78]. For example, bevacizumab is a monoclonal antibody that targets VEGF-A directly, preventing its ability to activate angiogenesis [79]. Another monoclonal antibody, 2C3, can bind to VEGF and selectively prevent its interaction with VEGFR-2 and not VEGFR-1 [78]. This selective inhibition has significantly reduced macrophage infiltration to the TME and metastasis in preclinical models of orthotopic breast and pancreatic cancer [80]. Typically, these drugs, such as bevacizumab, are given in combination with chemotherapy and have demonstrated long-term safety and manageable toxicity in their delivery [74,81]. In the phase III clinical trial (NCT00021060), the combination of bevacizumab with chemotherapy paclitaxel–carboplatin showed lower disease progression than chemotherapy-alone in NSCLC patients [82]. However, drug resistance often develops over time through normal intrinsic and acquired pathways but also due to tumors initiating alternative methods to cope with hypoxia [83].

A more recent approach to anti-angiogenic therapy is through the blockage of FGFR. Erdafitinib is a selective angiogenic tyrosine kinase inhibitor (TKI) that primarily targets FGFR-1-4 [84]. Erdafitinib reversibly inhibits FGFR kinase autophosphorylation to decrease downstream signaling and the activation of pathways such as RAS/MAPK/ERK and PI3K/AKT [84]. Erdafitinib’s current applications lie in treating metastatic and advanced urothelial cancer in patients carrying FGFR-2 or FGFR-3 gene mutations following platinum-based therapies [85]. Ongoing clinical trials are investigating the potential of erdafitinib in tumors with FGFR alternations, such as bladder cancer (phase IV, NCT05052372), NSCLC (phase II, NCT03827850), and advanced solid tumors (phase II, NCT04083976) [85]. 

Bemarituzumab is a recombinant humanized IgG2 monoclonal antibody demonstrating a high affinity for Fcγ receptor IIIa (FcγRIIIa) [86]. Bemarituzumab blocks the IgG III region of FGFR-2b, interfering with ligand binding and activating resultant downstream signaling pathways [86]. It also enhances antibody-dependent cellular cytotoxicity since FcγRIIIa-mediated ADCC activity is unique to NK cells [85].

**Table 1 biomedicines-11-02142-t001:** Summary of therapies.

Reference	Therapy	Type of Tumor	Efficacy of the Treatment
Haxho et al. [66]	oseltamivir phosphate	Mouse model of human pancreatic carcinoma, triple-negative breast adenocarcinoma, and human ovarian carcinoma	Mice treated with OP 50 mg/kg survived up to 4.5 times longer than those in untreated or OP 30 mg/kg treated groups.
Khoury et al. [76] Bałan et al. [77]	pentoxifylline	Mouse model of sarcoma	Transplantation of pentoxifylline-treated tumor cells results in a significant reduction in tumor size and number. Treatment has a dose-dependent effect on angiogenesis inhibition.
Hurwitz et al. [79]	bevacizumab	Metastatic colorectal cancer	Survival analyses show a hazard ratio of 0.66 (*p* < 0.001) for mortality and 0.54 (*p* < 0.001) for progression with the combined treatment of IFL with bevacizumab compared with IFL alone.
Roland et al. [80]	2C3	Preclinical models of orthotopic breast and pancreatic cancer	Treatment with 2C3 inhibits angiogenesis and macrophage infiltration in the tumor microenvironment. This significantly reduces tumor burden and metastatic activity.
Sandler et al. [82] (phase III clinical trial)	bevacizumab + PC	NSCLC	Survival analyses show a hazard ratio for disease progression of 0.66 (95% CI, 0.57 to 0.77; *p* < 0.001) with the combination of Bevacizumab and Carboplatin-Paclitaxel compared with chemotherapy-alone.
Siefker-Radtke et al. [84]	erdafitinib	Advanced urothelial cancer, bladder cancer, NSCLC, and advanced solid tumors	Treatment with a continuous dose of 8 mg or 8–9 mg Erdafitinib resulted in a median progression-free survival of 5.5 months (95% CI, 4.3 to 6.0).
Wainberg et al. [87]	bemarituzumab	HER2-negative gastroesophageal junction adenocarcinoma and gastric cancer	The hazard ratio of 0.68 (95% CI, 0.44 to 1.04; *p* = 0.073) from survival analyses comparing bemarituzumab-treated and placebo-treated groups was not significant.
**Reference**	**Therapy**	**Type of tumor**	**Efficacy of the treatment**
Cai et al. [88]	BGJ398 + rapamycin	Mouse model of ovarian cancer	Combined treatment with BGJ398 and rapamycin in a 1:1 ratio resulted in dose-dependent inhibition of cell proliferation. Inhibitory effects were observed on cell growth, motility, angiogenic marker expression, and associated proteins’ phosphorylation. The treatment induces cell cycle arrest and cell apoptosis, leading to a reduction in tumor size.
Staehler et al. [89] (phase IV clinical trial)	everolimus	Metastatic colorectal cancer	Treatment with everolimus after no response to VEGF inhibitors improved the survival of patients, with a 6-month PFS of 39.9%.
Cheng et al. [90]	atezolizumab + bevacizumab	Unresectable locally advanced or metastatic HCC	Combination vs. sorafenib-only: PFS 6.9 vs. 4.3 months; ORR 30% vs. 11%; OS 19.2 vs. 13.4 months
Motzer et al. [91] (phase III clinical trial)	avelumab + axitinib	Advanced renal-cell carcinoma	Survival analyses of patients with PD-L1-positive tumors showed a hazard ratio of 0.61 (95% CI, 0.47 to 0.79; *p* < 0.001) for disease progression or death with avelumab + axitinib versus sunitinib. ORR 55.2% (95% CI, 49.0 to 61.2) and 25.5% (95% CI, 20.6 to 30.9) among patients who received these agents as first-line treatment.

Abbreviations: HCC: Hepatocellular carcinoma; IFL: Irinotecan, Fluorouracil, and Leucovorin; mTOR: Mechanistic target of rapamycin; NSCLC: Non-Small Cell Lung Cancer; OP: Oseltamivir Phosphate; ORR: Objective response rate; OS: Overall survival; PC: paclitaxel–carboplatin; PD-L1: Programmed death-ligand 1; PFS: Progression-free Survival; TKI: Tyrosine Kinase inhibitor; VEGF-A: Vascular endothelial growth factor A; VEGF: Vascular endothelial growth factor; VEGFR-2: Vascular endothelial growth factor receptor 2.

An exploratory phase II trial of bemarituzumab (NCT03694522) in HER2-negative gastroesophageal junction adenocarcinomas and gastric cancer promising clinical efficacy was reported despite no statistically significant improvement in progression-free survival [87]. A phase III trial (NCT05052801) is currently investigating bemarituzumab in combination with mFOLFOX6 in gastric or gastroesophageal junction cancer with FGFR-2b overexpression [86]. Drug resistance has also been noted to develop over time to FGFR-blocking drugs. One approach to overcome this barrier is through combination therapy of mammalian targets of rapamycin (mTOR) and FGFR blockades [88]. mTOR is a serine/threonine kinase that stimulates the PI3K/AKT signaling pathway to drive angiogenesis and cell proliferation [85,88]. A study in ovarian cancer mouse models reported that BGJ398 and rapamycin inhibition of FGFR and mTOR reduced tumor size and induced tumor regression, apoptosis, and cell cycle arrest in OC cells [88]. Everolimus, an oral and more pharmacokinetically favorable analog of rapamycin, exhibited a prolonged progression-free survival rate of patients with advanced renal cell carcinoma (RCC) after VEGFR-2 inhibitors in a randomized, double-blind, phase III clinical trial (NCT00410124) [92]. More recently, a phase IV clinical trial (NCT01266837) found similar results, with a 6-month progression-free survival (PFS) of 39.3% in metastatic colorectal cancer patients treated with everolimus after VEGFR inhibitors [89].

### Immune Checkpoint Inhibitor and Anti-Angiogenesis Combination

Immune checkpoint inhibitors (ICIs) rejuvenate anergic and exhausted CTLs to exert anti-tumor effects [93,94]. Despite advances in ICIs, only 20-30% of cancer patients respond to ICIs alone, and the prediction of responders to treatment is subpar due to the immune system’s complexity and lack of a single predictive biomarker [93]. Anti-PD1, PD-L1, and CTLA-4 therapy prolong anti-angiogenic therapy, facilitating further CD4^+^ T-cell activation and vessel normalization [94,95]. The combination of anti-angiogenic and immune checkpoint inhibitor therapy has demonstrated a synergistic effect in cancer [93].

Allen et al. demonstrated that the combination of anti-angiogenic therapy targeting VEGFR-2 and anti-PD-L1 immunotherapy resulted in improved efficacy in models of pancreatic neuroendocrine tumor (RT2-PNET), mammary carcinoma (MMTV-PyMT) and glioblastoma (NFpp10-GBM) [93,94]. The anti-VEGFR-2 treatment increased the expression of PD-L1 in tumors through the secretion of IFN-γ by CD8^+^ T-cells, which can potentially increase the sensitivity of anti-PD-L1 therapy [93,94]. Additionally, the combination approach increased pericyte coverage and the normalization of tumor vessels, facilitating activated T-cell infiltration into the tumor microenvironment [93,94]. Anti-angiogenic therapies can promote the maturation and activation of dendritic cells leading to an improved presentation of tumor antigens to T-cells, enhancing the effectiveness of immune checkpoint inhibitors that target T-cell activation [93]. Anti-angiogenic therapies can also reduce the accumulation and activity of immunosuppressive cells, such as Tregs and MDSCs, within the tumor microenvironment. This reduction in immunosuppressive cells can create a more favorable immune environment for the immune checkpoint inhibitors to work effectively [93].

Combination therapy of ICI and anti-angiogenic agents has also demonstrated success in human clinical trials. Atezolizumab, an anti-PD-L1, when combined with bevacizumab, an anti-VEGF-A, significantly increased overall survival (OS) by 5.8 months, progression-free survival (PFS) by 2.6 months, and the objective response rate (ORR) by 19% relative to sorafenib in adult patients with unresectable locally advanced or metastatic hepatocellular carcinoma (HCC) [90]. Furthermore, the 18-month survival rate for the atezolizumab and bevacizumab combination compared to sorafenib alone is 52% and 40%, respectively [90]. This combination therapy is FDA-approved to treat HCC, and in combination with carboplatin and paclitaxel, it is approved as a first-line treatment for patients with NSCLC [96]. While atezolizumab and bevacizumab are the only ICI/anti-VEGF therapies that are FDA-approved, there are many promising clinical trials in phase II/III with other anti-angiogenic and ICI combinations. One example is the use of avelumab (anti-PD-L1) and axitinib (anti-VEGFR-2), which, when used in combination, show better ORR compared to patients treated with sunitinib (TKI inhibitor) in advanced renal-cell carcinoma [91].

It is essential to acknowledge that ICIs are mainly focused on anti-VEGF or anti-VEGFR agents; however, research should be diversified to other AFs such as FGF, PDGF, PIGF, and SDF-1α considering understanding the alternative pro-angiogenic pathways that contribute to anti-VEGF inhibitor resistance. Many ongoing clinical trials are investigating multi-target anti-angiogenic inhibitors and alternative angiogenic pathways, such as FGF, in combination with ICIs. One example is a phase II clinical trial (NCT03473743) investigating the co-inhibition of FGFR and PD-1 through co-administration of erdafitinib and cetrelimab in cisplatin-ineligible patients [97]. Furthermore, it is essential to continue research into reliable predictive biomarkers for anti-VEGF response, ICI response, and combination response.

## 5. Conclusions

Research surrounding angiogenesis is progressing in the right direction. There is growing evidence that angiogenesis is critical in cancer progression and that targeting the signaling pathways involved can have anti-tumor responses. Various therapeutic options are being studied, with combination therapy being suggested as the future in anti-angiogenic therapy. The tumor microenvironment is characterized by immunosuppression. ICIs promote anti-tumor action by the immune system and synergize with other inhibitory pathways, such as anti-angiogenic agents. However, much research is still needed before a gold treatment standard can be coined. Furthermore, understanding angiogenic drug resistance highlights the need for additional research to regulate alternative non-VEGF-related pro-angiogenic pathways.

## Figures and Tables

**Figure 1 biomedicines-11-02142-f001:**
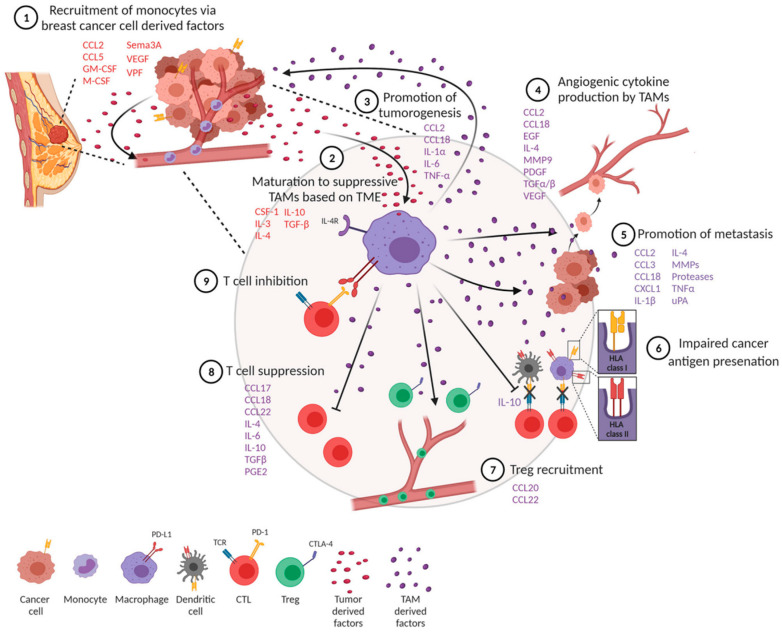
Overview of the role TAM polarization and cytokine/growth factor recruitment plays in breast cancer tumorigenesis, metastasis, and immune evasion. Citation: © 2021 Mehta, Kadel, Townsend, Oliwa and Guerriero. Frontiers in immunology 2021, 12:643,771, https://www.ncbi.nlm.nih.gov/pmc/articles/PMC8102870/ (accessed on 27 July 2023). This is an Open Access article that permits unrestricted non-commercial use, provided the original work is properly cited.

**Figure 2 biomedicines-11-02142-f002:**
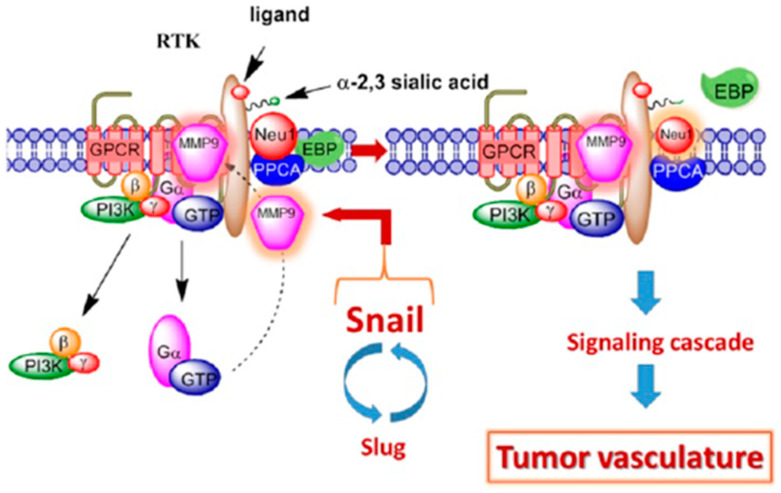
Snail and MMP-9 signaling axis in facilitating a neuraminidase-1 (Neu1) and matrix metalloproteinase-9 (MMP-9) crosstalk in regulating receptor tyrosine kinases (RTKs) in cancer cells to promote tumor neovascularization. Abbreviations: EBP: elastin binding protein; GPCR: G-protein coupled receptor; GTP: guanine triphosphate; Pi3K: phosphatidylinositol 3-kinase; PPCA: protective protein cathepsin A. Citation: Taken in part from Research and Reports in Biochemistry 2013:3 17–30. © 2013 Abdulkhalek et al., publisher and licensee Dove Medical Press Ltd. and Abdulkhalek et al. Clinical and Translational Medicine 2014, 3:28, http://www.clintransmed.com/content/3/1/28 (accessed on 27 July 2023). This is an Open Access article that permits unrestricted non-commercial use, provided the original work is properly cited.

## Data Availability

Data are contained within the article.

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
