# Peer review of "Metamorphic Effect of Angiogenic Switch in Tumor Development: Conundrum of Tumor Angiogenesis Toward Progression and Metastatic Potential"

_biomedicines, 2023, doi:10.3390/biomedicines11082142_

Round 1

Reviewer 1 Report

Dear authors,

Thanks for your contribution on this field. This is an interesting state of art on angiogenic growth factors and more precisely on their role during tumor development beyond their classic roles in blood vessels formation. All the aspects are almost cover. The manuscript is still well written but not well organized.

Indeed, all these new roles are included in in the introduction part. To my point of view, the section 1.2 should represent a section on its own, so section 2. (Line 73)

This review is limited at only one figure which is quite surprising since it can be expected to have at least 2-3 figures for a review.

Moreover, the only figure is coming from an article published in 2013.

It can be expected that new data should have come out in 10 years’ time!

Please modified this figure by upgrading with new knowledge.

Other figures highlighting new findings within the different sections.

Minor points:

Line 14: please remove references from abstract.

Line 214: indeed, it is proMMP-9 instead of MMP9 since its activation is evocated.

Lines 215-217; 225: if the overexpression is at protein level, then it should be MMP-9 and MMP-7.

Lines 244-245: here it is a transcriptional regulation so it should be mmp9 (small letter in italic without dash)

Lines 247-249: please provide a reference for this statement.

All the best,

Author Response

Reviewer # 1

Comments and Suggestions for Authors

Dear authors,

Thanks for your contribution on this field. This is an interesting state of art on angiogenic growth factors and more precisely on their role during tumor development beyond their classic roles in blood vessels formation. All the aspects are almost cover. The manuscript is still well written but not well organized.

Indeed, all these new roles are included in in the introduction part. To my point of view, the section 1.2 should represent a section on its own, so section 2. (Line 73)

Thank you for the feedback. We have updated it so that “Contribution of angiogenic growth factors in immunosuppression and immune escape” is a section of its own.

This review is limited at only one figure which is quite surprising since it can be expected to have at least 2-3 figures for a review.

Moreover, the only figure is coming from an article published in 2013.

It can be expected that new data should have come out in 10 years’ time!

Please modified this figure by upgrading with new knowledge.

Other figures highlighting new findings within the different sections.

A figure highlighting TAMs in tumorigenesis/metastasis and recruitment of growth factors was added (See figure 1). This figure was from an open access paper published in 2021 (Mehta, Kadel, Townsend, Oliwa and Guerriero. Frontiers in immunology 2021, 12:643771, https://www.ncbi.nlm.nih.gov/pmc/articles/PMC8102870/)

Minor points:

Line 14: please remove references from abstract.

Line 214: indeed, it is proMMP-9 instead of MMP9 since its activation is evocated.

Lines 215-217; 225: if the overexpression is at protein level, then it should be MMP-9 and MMP-7.

Lines 244-245: here it is a transcriptional regulation so it should be mmp9 (small letter in italic without dash)

Lines 247-249: please provide a reference for this statement.

Thank you for the comments. We removed the references from the abstract and moved them to the introduction.

Regarding line 214, the paper cited only mentions MMP9 and not proMMP-9, therefore I have left it as is. Please see: Zhang et al., 2015. doi:10.1159/000430244 for further clarification.

For consistency in the paper, we will be using MMP-9, except for lines 244-245 which describes the transcriptional regulation (now presented in small letters in italic without dash).

Reference for lines 247-249 was added (Abdulkhalek et al., 2014. doi:10.1186/s40169-014-0028-z.).

All the best,

 Submission Date

29 June 2023

Date of this review

17 Jul 2023 17:20:58

Reviewer 2 Report

This manuscript reviewed the current findings of angiogenic growth factors in tumor development and anti-angiogenic treatments in cancer therapy. The topic is interested. However, there are some minor issues to consider.

l   I would suggest to remove the cited references in the abstract.

l   Line 149. “CD4+T cell” should be “CD4+ T cell”.

l   Line 156. “CD4+” should be “CD4+”.

l   Line 157. “CD8+ T cells” should be “CD8+ T cells”.

l   Line 173. “M2TAMs” should be “M2-TAMs”.

l   Line 181. “CD8+ T-cell” should be “CD8+ T cell”.

l   Line 186. “CD8+ T-cells” should be “CD8+ T cells”.

l   Line 187. “CD8+ and CD4+ T-cell” should be “CD8+ and CD4+ T cell”.

l   Line 190. “T-regs” should be “Tregs”.

l   Line 191. I would suggest to revise as follows:”…also increased the survival of Tregs through IL-2-mediated….”.

l   Line 241, 244, 247, 259, 261, 264, 283, 284. “MMP-9” should be “MMP9”

l   Line 351. “CD4+” should be “CD4+”.

l   Line 358.” “CD8+ T cells” should be “CD8+ T cells”

I would suggest the authors discuss what angiogenesis inhibitors are being used to treat cancer in humans in the manuscript.

Author Response

Reviewer #2

Comments and Suggestions for Authors

This manuscript reviewed the current findings of angiogenic growth factors in tumor development and anti-angiogenic treatments in cancer therapy. The topic is interested. However, there are some minor issues to consider.

 l   I would suggest to remove the cited references in the abstract.

l   Line 149. “CD4+T cell” should be “CD4+ T cell”.

l   Line 156. “CD4+” should be “CD4+”.

l   Line 157. “CD8+ T cells” should be “CD8+ T cells”.

l   Line 173. “M2TAMs” should be “M2-TAMs”.

l   Line 181. “CD8+ T-cell” should be “CD8+ T cell”.

l   Line 186. “CD8+ T-cells” should be “CD8+ T cells”.

l   Line 187. “CD8+ and CD4+ T-cell” should be “CD8+ and CD4+ T cell”.

l   Line 190. “T-regs” should be “Tregs”.

l   Line 191. I would suggest to revise as follows:”…also increased the survival of Tregs through IL-2-mediated….”.

l   Line 241, 244, 247, 259, 261, 264, 283, 284. “MMP-9” should be “MMP9”

l   Line 351. “CD4+” should be “CD4+”.

l   Line 358.” “CD8+ T cells” should be “CD8+ T cells”.

Thank you for the comments. We removed the references from the abstract and moved them to the introduction. Here are the corrections we made to maintain consistency throughout the paper:

  • All “CD4+” and “CD8+” have been changed to have the + be superscripted (now: CD4+, CD8+)
  • All “T cells” have been changed to include the dash (now: T-cell)
  • All “T-regs” have been changed to not include the dash (now: Treg)
  • All “M2TAMs” have been changed to include the dash (now: M2-TAMs)

Line 191 has also been corrected using your suggestion.

Regarding the MMP-9 comments (Line 241, 244, 247, 259, 261, 264, 283, 284): Reviewer 1 commented that MMP9 should be MMP-9. A look through the literature found that both forms are used interchangeably. For consistency in our paper, we will be using MMP-9.

I would suggest the authors discuss what angiogenesis inhibitors are being used to treat cancer in humans in the manuscript.

We also added multiple clinical trial studies looking at angiogenesis inhibitors to treat cancers in humans.

Submission Date

29 June 2023

Date of this review

13 Jul 2023 22:02:20

Round 2

Reviewer 1 Report

Dear authors,

Thanks for considering my comments.

A last modification, please: remove "Graphical abstract of " in line 314 ;-)

Congrats for your contribution

All the best

Author Response

Thanks for considering my comments.

A last modification, please: remove "Graphical abstract of " in line 314 ;-)

Author response: We removed "graphical abstract of" in the figure legend. Thank you for your excellent review of the manuscript.

Congrats for your contribution

All the best